# Critical time intervention for people leaving prison at risk of homelessness in England and Wales (PHaCT trial): a pilot feasibility randomised controlled trial

Adam Dale Newman Williams [iD],[1] Nina Jacob [iD],[2] Detelina Grozeva [iD],[1] Barry Lloyd,[1] Yvonne Moriarty [iD],[2] Manuela Deidda [iD],[3] Samuel Owusu Achiaw,[4] Ian Thomas,[1] Kelly Lewis,[1] Rebecca Cannings-John [iD],[2] Iolo Madoc-Jones,[5] Suzanne Fitzpatrick,[6] Srinivasa Vittal Katikireddi [iD],[7] Peter Mackie,[8] James White,[2] Jim Lewsey [iD] [4]

► https://doi.org/10.1136/bmjopen-2024-097761

For numbered affiliations see end of article.

**Correspondence to**
Adam Dale Newman Williams;
williamsAD7@cardiff.ac.uk

## ABSTRACT

**Objective** To determine whether a full-scale randomised control trial (RCT) assessing the efficacy and cost-effectiveness of a housing led Critical Time Intervention (CTI) is feasible and acceptable.

**Design** Pilot parallel two-arm individual level RCT, including process evaluation and embedded exploratory health economic evaluation.

**Setting** Four prisons for men across England and Wales, UK.

**Participants** Men leaving prison at risk of homelessness and intervention delivery staff.

**Intervention** CTI has four components: (1) pre-engagement phase: assessing the needs of the client and implementing a plan pre-discharge; (2) transition to community: forming relationships and goal setting; (3) try out: encouraging problem-solving and managing practical issues and (4) transfer of care: developing long-term goals and transferring responsibilities to community providers.

**Outcome measures** Progression criteria: recruitment, retention, acceptability of the processes (CTI and trial method) and fidelity of intervention delivery. We also assessed the completeness of primary, secondary and exploratory outcome measures and estimated intervention costs.

**Results** The recruitment progression criterion was met, with 92% (34/37) of approached individuals consenting to participate (target: 50%). However, the overall recruitment target of 80 was not achieved, and retention was low, only 18% (6/34) provided follow-up data, well below the 60% threshold. Retention was hindered by systemic challenges, including changes to prison release policies and reduced probation support. While the CTI model was acceptable to staff and service users, the trial design, particularly randomisation, was not. Intervention fidelity met the progression criteria. Baseline data collection for health economics and resource use was feasible, and intervention costs were estimated.

**Conclusion** This pilot trial identified significant challenges to conducting a full-scale RCT of CTI in this context, particularly around retention, trial acceptability and systemic instability. While CTI remains a promising model, a traditional RCT design may not be viable in this setting without substantial structural and ethical adaptations.

**Trial registration number** ISRCTN46969988.

## STRENGTHS AND LIMITATIONS OF THIS STUDY

⇒ The study used a pilot parallel two-arm randomised controlled trial with embedded process and exploratory economic evaluations, offering a robust framework for feasibility assessment.

⇒ Ethical and practical concerns around randomisation affected trial implementation and acceptability.

⇒ Logistical challenges, including delays in approvals and difficulties maintaining contact post-release, impacted trial delivery.

⇒ External factors, such as changes in prison resettlement policies and staffing shortages, posed major barriers to recruitment and retention.

## INTRODUCTION

For people leaving prison, re-integrating into the community can be a difficult process. The absence of suitable affordable housing and a stable support network, in the context that many may have complex needs linked to substance use and mental ill-health, makes people leaving prison more susceptible to homelessness.[1] The absence of stable accommodation can then have further repercussions. People experiencing homelessness are at a higher risk than the general population of infectious and non-communicable diseases, mental health problems, alcohol and substance use, have higher rates of hospital admissions and report lower levels

of well-being and health-related quality of life.[1–4] Health outcomes are substantially worse among people with a history of both imprisonment and homelessness than each in isolation.[5] The detrimental consequences of homelessness following imprisonment can lead to a huge societal burden in terms of healthcare and societal costs.[6]

Critical Time Interventions (CTIs) are structured, time-limited case management approaches designed to support individuals during periods of transition, such as release from prison, by enhancing engagement with treatment and community services through problem-solving and continuity of care.[7] Core components include small caseloads, active outreach, personalised case management, psychosocial skill-building and motivational support, delivered through a phased model that tapers in intensity over time.[7 8] A multisite randomised control trial (RCT) in the UK demonstrated that CTI could be feasibly and acceptably implemented in prison settings, showing improved service engagement and providing valuable economic evaluation data.[8] While international evidence suggests CTI may improve housing stability among people experiencing or at risk of homelessness, its impact on health outcomes and quality of life remains limited.[8–12] Gaps remain, particularly the limited UK-based RCTs among prison-leavers facing housing instability, highlighting the need for further research in this area. The overall aim was to conduct a pilot RCT to determine whether a future full-scale RCT assessing effectiveness and cost-effectiveness of a housing-led CTI is warranted.

## METHODS
The full protocol is available in the online supplemental material.

### Progression criteria
We aimed to recruit 20 participants from each of the four prisons in which the intervention was being delivered, totalling 80 participants. The sample size of 80 participants at the end of the pilot would allow binary outcomes to be estimated. A traffic light system was in place to judge feasibility:

1. Recruitment: 50% of those approached agree to participate.
2. Retention: 60% of those recruited, in both intervention and control groups, are retained at final follow-up.
3. The process evaluation provides evidence that the process is acceptable for participants and staff delivering the intervention.
4. The intervention is implemented with fidelity in all settings.

### Trial design
This pilot study undertook a parallel two-arm, individual level RCT of a pre-existing CTI intervention, including process evaluation and embedded exploratory health economic evaluation.

### Setting
Participants were intended to be recruited from four prisons across England and Wales, housing adult males. The locations were predetermined by where the intervention was already being delivered by the intervention provider (CTI team). During recruitment, the protocol was updated to include an additional site, but this site did not progress to recruitment. At completion, only three of the four prisons provided participants.

### Participants
Inclusion criteria:
- ► Men leaving prison at risk of homelessness.
- ► Aged over 18 years.
- ► Released into the local authority areas that CTI is being delivered and have a local connection (eg, lived there previously; close relatives living in these areas).
- ► Recourse to public funds.
- ► Have experienced prison and homelessness at least once.
- ► Have mental health or substance use support needs.

Exclusion criteria:
- ► Under the Multi-Agency Public Protection Arrangements panel 3.
- ► In receipt of Housing First (ie, support needs too high/complex to benefit from CTI).

### Randomisation and blinding
Randomisation was completed after baseline data collection. Participants were randomised on a 1:1 ratio and stratified by site, using random block sizes. Randomisation was completed using concealed paper envelopes. Envelopes contained the arm allocation and participant identification number. Boxes and envelopes were clearly labelled with the prison name and sequence number. Training was provided to recruitment staff to outline the randomisation method. Participants were randomised from October 2023 to July 2024.

### Intervention
Figure 1 shows the logic model for CTI, a psychosocial approach supporting individuals transitioning from institutions to the community. This adaptation from the UK follows a 'housing-led' model, prioritising rapid access to stable housing as a foundation for addressing broader health and social needs.

CTI strengthens long-term connections to services, family and peers through time-limited emotional and practical support. Clients co-develop personal plans with goals for short, medium and long term to build self-management skills, empowerment and improve well-being. Stable housing is central, reducing the stress and instability linked to homelessness.

The intervention begins before discharge, with an assessment panel identifying individuals at risk of homelessness. Caseworkers initiate engagement, assess needs and co-create a tailored transition plan. Post release,

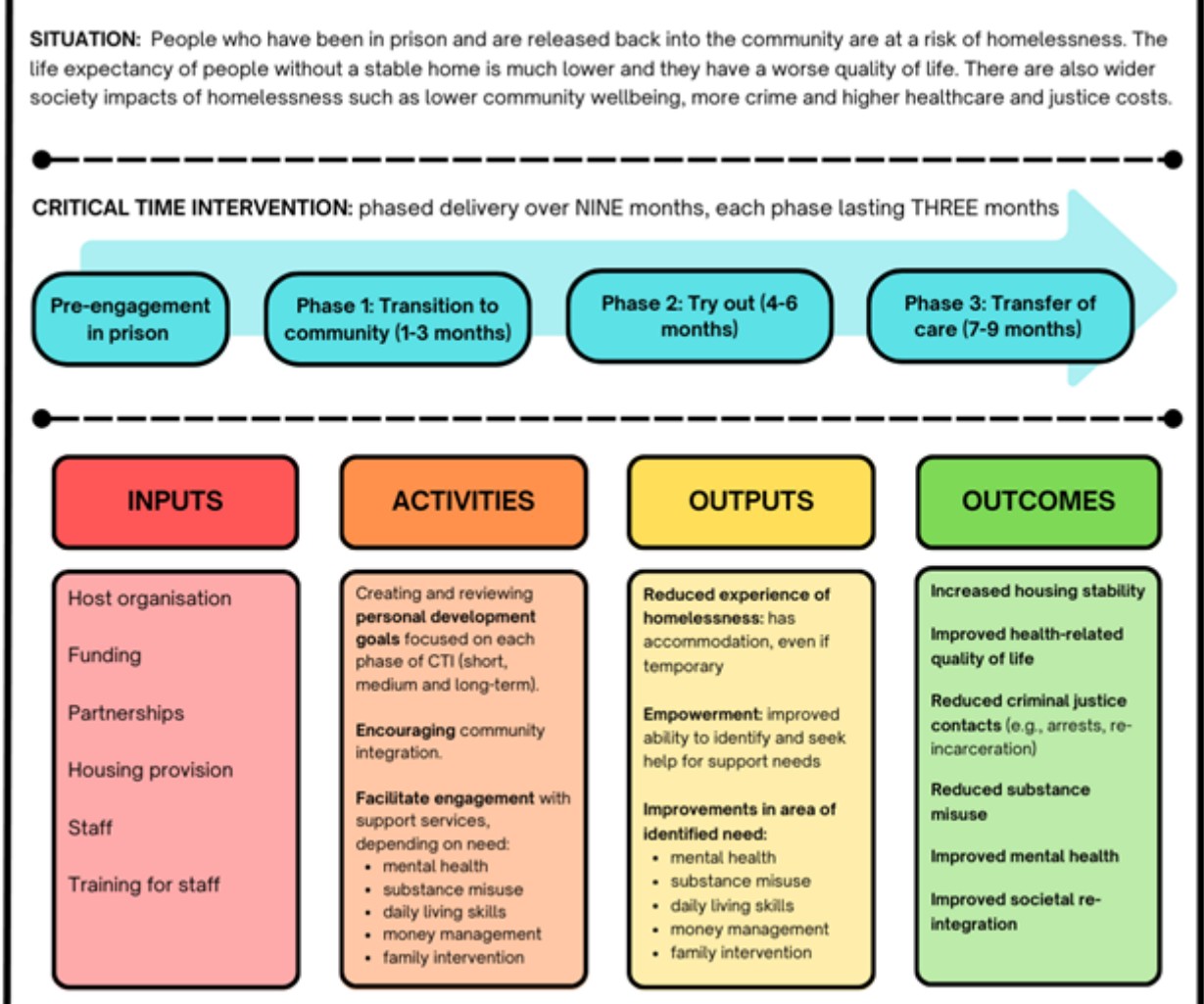

**Figure 1** Housing-led CTI logic model. CTI, Critical Time Intervention.

clients are supported into tenancies and progress through three phases:

► *Phase 1: transition to community (3 months)*—intensive support, weekly visits, goal-setting and crisis resolution.
► *Phase 2: try out (3 months)*—reduced contact; clients practise independence with support adjusted as needed.
► *Phase 3: transfer of care (3 months)*—caseworker steps back, finalising handover to community supports and caregivers.

CTI was delivered by caseworkers from a charity in the UK specialising in housing support for prison leavers. Support was provided in prisons, offices, community settings and residences across England and Wales. The intervention was funded by a charity in the UK.

### Ethics and consent

The study received NHS ethics approval from Wales REC 3 (22/WA/0347), the National Research Council (NRC) approval (which governs research within prison and probation settings), and permissions from each prison Governor and Probation Regional Manager. The study

was registered in ISRCTN46969988. Access to routine data via the Secure Anonymised Information Linkage (SAIL) Databank was granted by the SAIL Information Governance Review Panel (SAIL ID:1365).

Potential participants were referred to the CTI team for eligibility screening by the Prison, Community Offender Management Team or a self-referral. The CTI team would engage the individual and complete their referral assessment. Those deemed eligible were provided with a brief verbal explanation of the trial with the support of a summary information sheet and asked if they were interested in taking part. For those interested, prison visits were organised for the study to be explained, information sheets and consent forms to be signed and, following informed consent, baseline data collection occurred.

### Data collection
#### Trial data collection

The CTI team identified eligible individuals based at prisons where they delivered the intervention. At recruitment, baseline data collection occurred in the form of

paper surveys in the prisons where participants were located.

Follow-up was scheduled at 3, 6, 9 and 12 months post-randomisation, with a 2-week flexible window around the scheduled time point to accommodate participant availability and logistical constraints. Follow-ups were conducted in probation offices, intervention team offices or via telephone. Probation services were engaged to help track participants post-release and facilitate follow-up contacts, although their involvement varied by site and availability.

The intention was to use clinical research nurses (CRNs, registered nurses specialised in managing and conducting clinical trials and research studies) to complete recruitment, baseline data and follow-up data collection for England sites with the local Research and Development (R&D) delivery team (the Welsh equivalent of CRN) being used for Wales. The use of CRNs is standard in health research and was chosen for this study to reflect the approach that would be adopted in a full-scale UK-wide trial. In Wales, recruitment, baseline data collection and follow-up were completed by the local R&D delivery team, while recruitment and data collection in England were completed by the research team due to logistical challenges with the CRN teams.

### Process evaluation

Semi-structured interviews were to be conducted with participants and CTI caseworkers from both England and Wales. The intention was to conduct 24 interviews with participants from the intervention group (12 participants to be interviewed at phase 1 and 3 of the intervention) and 16 interviews with participants from the control group.

Twenty-four observations were planned with service users of the intervention, twelve with participants in the RCT control group and twelve with service users of the CTI model who started receiving support before the study randomisation began. These ranged across the phases of the intervention. A full description of the process evaluation is presented in the following paper.[13]

### Health economic evaluation

Cost logs from the intervention delivery team examined time and resources spent implementing the intervention.

### Outcome measures
#### Feasibility outcomes

These criteria sought to assess the number who agreed to participate, the retention at the end of the study, the acceptability of the delivery of the intervention and RCT design and fidelity of delivery, with thresholds set according to a traffic light system.

#### Trial outcome measures

The indicative primary outcome for the trial was health-related quality of life, measured using the validated 12-item short form health survey (SF-12).[14] This was chosen for being a recognised and validated outcome in public health and social care research, making it a meaningful and policy-relevant indicator for assessing the broader benefits of CTI beyond specific behavioural or clinical endpoints. The secondary outcomes are displayed in table 1 and include housing stability, capability well-being, hazardous alcohol consumption and substance use, and resource use and mortality.

The exploratory outcomes assessed whether participants consented to link to routinely collected healthcare and criminal justice data. Of those who consented, we also sought to explore the proportion of people who could be successfully linked to their healthcare records, via SAIL/NHS England, and criminal justice records, via the Ministry of Justice Data Lab, by applying to use these data linkage services. Attempting 'test linkages' sought to inform the potential use of data linkage to provide additional secondary outcomes, such as re-incarceration, re-conviction, education, training and employment, and welfare benefits receipt.

### Analysis

No power calculation was performed as our aim was to evaluate acceptability of the intervention, not to estimate effectiveness. This study determined rates of recruitment, response and retention, and the distribution of the indicative primary outcome measures to inform a full-scale trial sample size calculation. Given there was capacity to deliver the intervention to 120 participants per year, we aimed to recruit 80 participants (40 per arm) over 4 months allowing us to estimate a recruitment rate of 50% within 95% CI±11% (75% rate±9.5%).

### Statistical analysis

The primary analysis sought to determine whether the pre-specified progression criteria were met. The analyses were primarily descriptive, providing estimates of recruitment, response and retention rates. Recruitment, randomisation and retention of participants at follow-up were summarised in a Consolidated Standards of Reporting Trials (CONSORT) flow diagram (figure 2). We tabulated demographic characteristics of participants within settings by study arm (intervention or control) and assessment time point (baseline or follow-up) using descriptive statistics means and SDs (or medians and IQRs, as appropriate) for continuous outcomes, and frequencies and percentages for discrete outcomes. Analysis was performed in Stata V.18.[15]

### Qualitative analysis

Qualitative data generated through semi-structured interviews were audio-recorded, transcribed and coded. Field notes from observations and free-text entries in logbooks were coded using a similar system. Members of the research team analysed the data using inductive and deductive thematic analysis.[16] A coding scheme was developed by the researchers. The coding scheme evolved during analysis, with the new codes discussed and confirmed by the team, before being applied to

**Table 1**  Indicative and exploratory outcome measures

| Outcome | Measurement | Time point |
|---|---|---|
| **Indicative primary and secondary outcomes** | | |
| Health-related quality of life (primary outcome for full-scale trial) | SF-12[14] | Baseline Follow-up at 3, 6, 9 and 12 months |
| Health-related quality of life | EQ-5D: a standardised measure of health-related quality of life[21] | |
| Housing stability | ► Type of accommodation ► Number of days in a stable accommodation | |
| Substance use and hazardous alcohol consumption | AUDIT-C[22] | |
| Mortality | Recording participant deaths, including cause of death | |
| Capability well-being | ICECAP-A[21] | |
| Resource use | Ad hoc measurement of resource use for health economic evaluation | |
| **Exploratory outcomes** | | |
| Data linkage | Percentage of participants who consent to link their data anonymously with routine electronic data through SAIL, NHS England and Ministry of Justice Data Lab | Single point of data linkage |
| | Percentage of records successfully linked with data held in SAIL, NHS England and Ministry of Justice Data Lab to inform: ► Health and well-being ► Re-incarceration ► Crime records | |

AUDIT-C, Alcohol Use Disorders Identification Test Consumption; EQ-5D, EuroQol 5 Dimensions; ICECAP-A, ICEpop CAPability measure for Adults; SAIL, Secure Anonymised Information Linkage; SF-12, 12-item short form health survey.

previously coded data. NVivo V.12 supported data storage and analysis.[17]

Fidelity to the CTI model was measured against five fidelity items: housing-led approach, time-limited and phased approach, caseloads and supervision, person-centred approach/community focused, and harm reduction/recovery orientated approach. From the interviews and observations, a score was provided: (low=1, medium=2, high=3) against each fidelity item with an overall score calculated.

The qualitative findings will only be briefly mentioned in relation to the feasibility criteria. A full explanation of the findings of the process evaluation is available in the associated article.

### Economic evaluation

We reported descriptive statistics for health economics outcomes (EuroQol 5 Dimensions (EQ-5D) 3 Level and ICEpop CAPability measure for Adults (ICECAP-A)) and resource use by collection time point and intervention group. Quality of life and capability well-being scores were compared with reference values for the general population. The cost of delivering the CTI intervention was captured by adapting a tailored cost log to the existing data collection system used by the charity delivering the intervention.

### Patient and public involvement

An individual with lived experience of homelessness contributed to the design phase when applying for funding. The research questions were developed to align with the needs and experiences of the population at risk of homelessness.

### Challenges experienced

The study experienced significant delays that impacted recruitment timelines, beginning with approval and contractual issues. Although originally planned to open within 6 months, in Wales recruitment began after 16 months and after 22 months in England. Delays were caused by a prolonged NRC approval process and challenges in obtaining accurate contact details for relevant His Majesty's Prison and Probation Service (HMPPS) staff. The CTI team in England also faced staff shortages, which paused intervention recruitment for an extended period.

Additional complications arose with CRN teams. Although initially assigned to manage recruitment and follow-up, unexpected site-specific contractual

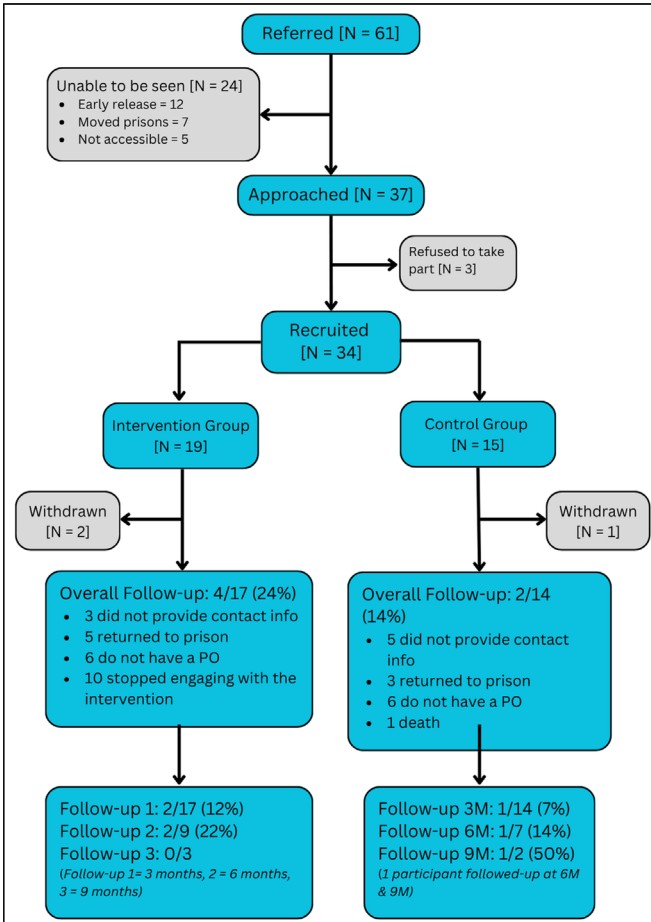

**Figure 2** PHaCT CONSORT diagram. *The denominator in the fractions with respect to the follow-up at 3, 6 and 9 months reflects the number of people who had the opportunity to be followed up in the lifetime of the study. CONSORT, Consolidated Standards of Reporting Trials; PO, probation officer.

requirements in the prison setting delayed their involvement. For recruitment in England (which was set to recruit 60 of the 80 participants) these delays led to a protocol amendment allowing the research team to take over recruitment and follow-up. However, as the team was

not locally based, this required intensive planning and clustered visits, which limited opportunities for participant engagement.

## RESULTS

In total, 34 individuals were recruited to the trial from an intended 80 participants (that is, 43% recruited from the target of 80 trial participants). Table 2 presents the recruitment figures across the study components. Fifteen individuals (44%) were randomised to the control arm and 19 (56%) to the intervention arm (figure 2, CONSORT diagram). Of the trial participants, the mean age was 38.1 years (SD: 11.2 years, min: 19, max: 67), and 9% were from ethnic minority backgrounds, comprising mixed or multiple ethnicities, Bangladeshi, and Caribbean or Black. Overall, 35% (n=12) reported not having any formal educational qualifications. Illicit drug use in the period of 12 months before entering prison was reported by 91% (n=31) of participants. Participants reported experiencing homelessness on average three times since the age of 16 years (IQR: 2–4), with 74% (n=25) reporting they were not in settled accommodation before entering prison. Full demographic information is available in the online supplemental material.

### Progression criteria

Table 3 summarises the results against the progression criteria. Two of the four progression criteria were rated green (recruitment and fidelity). The process evaluation was rated amber, with retention receiving a red rating.

### Recruitment

CTI caseworkers referred 61 people to the research team but 24 people were unable to be approached within the prison setting (so could not be recruited); reasons included early release (12/24, 50%), moving prison (7/24, 29%), not being accessible within the prison, that is, held in cell for misbehaviour, attending other meetings/prison visit (5/24, 21%). Of the 37 approached, 34 (92%) agreed to take part (the 'green' threshold for this criterion was ≥50%); for the three who refused, reasons

| Table 2 Summary of recruitment across study | | |
|---|---|---|
| | **Wales, site 1** | **England, site 2** |
| Intervention delivery team | CTI team | CTI team |
| Data collection team | R&D delivery team | Research team |
| Number of prisons providing participants | 1 | 2 |
| Trial participants recruited/ planned recruitment | 20/20 | 14/60 |
| Trial participants followed up | 5 | 1 |
| Number of interviews conducted (participants and staff) | 4 | 4 |
| Observation days | 10 | 20 |
| Observation locations | CTI office | CTI office, prison, probation and community settings |

CTI, Critical Time Intervention; R&D, research and development team.

**Table 3** Summary of results against the progression criteria

| Progression criterion | Red | Amber | Green | Actual |
|---|---|---|---|---|
| 50% of those approached agree to participate. | <40% | 40–49% | ≥50% | 92% |
| 60% of those recruited are retained at final follow-up. | <50% | 50–59% | ≥60% | 18% |
| The process (CTI and trial process) is acceptable for participants and staff. | Neither is accepted by participant or staff | CTI or trial process is accepted by participant or staff | Both are accepted by participant and staff | CTI=acceptable Trial=not acceptable |
| The intervention is implemented with fidelity in all settings. | Fidelity score <20 | Fidelity score 20–25 | Fidelity score ≥25 | Overall score=26 |

CTI, Critical Time Intervention.

included distrust of research and experiencing mental health difficulties. While the green threshold was met based on the wording of the criteria, the planned sample size of 80 participants was not achieved.

**Retention**

Of the 34 participants recruited, there was a death which occurred before follow-up, three withdrawals and eight participants returned to prison, leaving 22 people to attempt follow-up. The eight participants who returned to prison could not be followed up as it was either unknown which prison they had been sent to, or it was not a prison where we had approvals to conduct research. Six of the 34 recruited participants had follow-up data for at least one time point (18%, the 'green' threshold was 60%), two participants were allocated to the control group and four to the intervention. At 3 months, three participants were followed up (9%, 3/34), and at 6 months, three participants were followed up (9%, 3/34). One participant was followed up at 9 months (3%, 1/34). One participant was followed up for two time points; the rest were followed up at a single time point. Due to delays at trial initiation, no participants achieved the 12-month follow-up point. Of the 22 participants who could be followed up, eight (8/22, 36%) did not provide any contact details (due to not having any contact information or not knowing their details from memory). Those who did provide contact details (email, telephone or both) were contacted four times at each follow-up point. For many of the participants who provided a telephone number, it had been disconnected when contact was attempted (8/14, 57%).

To support follow-up, we would engage with probation officers to maintain contact with participants at follow-up; however, details of probation officers were not always known. Twelve (12/34, 35%) did not have any details for their probation practitioner at baseline, 12/34 (35%) had only a name with a partial address, and 10 participants (30%) had a name and contact information for their probation practitioner. Throughout the study period, the strain on the probation service following on from the End of Custody Supervised Licence (ECSL) and Probation

Reset scheme resulted in many of our participants not having contact with probation staff.[18]

**Process evaluation**

A detailed reporting of the findings from the process evaluation and fidelity assessment is provided in the associated paper. Only a summary is reported here for context.

The process evaluation aimed to assess the acceptability of the CTI model and the trial methodology, to both participants and staff delivering the intervention. In summary, the acceptability of the intervention was primarily reflected through the positive feedback and success stories shared by CTI caseworkers, as well as observational data indicating high acceptance among service users. The acceptability of the trial design was challenged by concerns about randomisation and equipoise, with some staff viewing randomisation as unethical due to limited support for vulnerable populations. This led to an amber rating.

**Fidelity**

Table 4 presents the summary of the fidelity scores across sites. Across the five domains, the intervention was assessed to be delivered with fidelity and scored to be green on the progression criteria.

Fidelity to the housing-led, time-limited model of CTI was largely maintained, with stable housing prioritised before addressing other needs. However, limited housing availability, especially for single men leaving prison, often delayed progress, leading to occasional extensions beyond the intended 9 months. Caseloads remained within or close to the recommended range, with site 1 averaging 15 and site 2 ranging from 18 to 28. This was considered manageable due to staff experience and phased support intensity. CTI was delivered with a strong person-centred focus, supporting service users to set goals and navigate complex systems. Harm reduction and recovery principles were embedded throughout, though unsuitable or temporary housing placements often compromised recovery efforts, requiring caseworkers to intensify support and connect individuals with relevant services.

**Table 4** Fidelity scores across sites

|  | Wales, site 1 | England, site 2 | Overall |
|---|---|---|---|
| Housing-led approach | 2 | 3 | 5 |
| Time-limited and phased approach | 2 | 2 | 4 |
| Caseloads and supervision | 3 | 2 | 5 |
| Person-centred approach/community focused | 3 | 3 | 6 |
| Harm reduction/recovery-orientated approach | 3 | 3 | 6 |
| Overall | **13** | **13** | **26** |

From the interviews and observations, a score was provided: (low=1, medium=2, high=3) against each fidelity item with an overall score calculated.

### Indicative outcomes

A breakdown of the indicative outcomes is available in the online supplemental material. The planned economic evaluation could not be conducted due to limited follow-up data. At baseline, the average EQ-5D score was 0.57 (SD=0.42) and ICECAP-A was 0.62 (SD=0.29). Most participants (73.5%) reported not being in settled accommodation before prison, with higher rates in the control group. Among six participants with follow-up data, housing situations varied, and two had returned to prison. Control group participants reported higher alcohol use prior to prison. While only two participants completed all CTI phases, missing data were minimal, supporting the feasibility of using SF-12, EQ-5D, ICECAP-A and resource use tools at baseline. Full details are provided in the online supplemental material.

### Exploratory outcomes

All participants consented to data linkage to healthcare data sources, and to link to criminal justice data sources. Due to delays in recruitment, and as most of the sample were initially recruited from the Welsh study setting, test linkages using NHS England were not conducted. The intervention delivery organisation attempted to access the Justice Data Lab but was unable to use the service due to issues around the timings of data collection as not enough time had elapsed for 1 year offending to have taken place following exit from the CTI intervention. Test linkages were, however, attempted via the SAIL Databank. As SAIL only relates to people who have been resident in Wales, people recruited from sites in England were excluded from the test linkage (14/34, 41%). In total, data relating to 17 participants at baseline were sent to SAIL for matching. Viable linkages were possible for 11 participants (65%) whose information was sent to SAIL, with the remaining 6 participants (35%) having non-viable linkages, either due to no link being possible or because the probabilistic matching threshold was less than 90%.

### Economic evaluation

Due to the small number of participants completing follow-up, it was not possible to conduct the planned economic evaluation or compare outcomes and costs over time. Instead, the study presents a descriptive analysis of CTI intervention costs, which totalled £6356, with a detailed breakdown provided in the online supplemental material. Only 2 of the 17 participants randomised to CTI completed all intervention stages. Cost estimates should be interpreted cautiously due to limited data and missing travel cost components.

### DISCUSSION

This pilot RCT assessed the feasibility of delivering a housing-led CTI for prison-leavers experiencing homelessness. Of the four pre-specified progression criteria, only two were met, suggesting that a full-scale RCT of this model, in its current form, is not feasible. While agreement to participate was high (92% of those approached consented), we did not achieve our intended sample and retention was low, with only 18% of participants being contactable for follow-up. This was largely due to contextual disruptions, including the introduction of the End of Custody Supervised Licence scheme, which accelerated release timelines and undermined recruitment and follow-up processes.

Compared with previous studies, such as the UK-based RCT of CTI by Shaw et al,[8] which demonstrated feasibility and positive outcomes in service engagement, our trial faced more significant structural and operational barriers. These included rapid policy changes, high licence recall rates and staffing shortages in probation services, all of which contributed to instability and reduced participant engagement.[18–20] While quasi-experimental designs are a valid alternative, they are less robust than RCTs for establishing intervention effectiveness. Importantly, RCTs are feasible in prison settings, as demonstrated in previous studies,[8] but they require embedded research teams and stable operational conditions. These conditions were not consistently present in this study, which followed a traditional health research model using external partners for data collection managed by a central trial team, an approach that proved less suited to the dynamic and restrictive prison environment.

A key limitation was the inability to offer incentives due to NRC restrictions. This policy affected all participants,

not just those in the control group, and likely contributed to low retention. The lack of compensation, standard in general population research, raises ethical concerns around equity and fairness, particularly when working with marginalised groups. Alternative engagement strategies, such as maintaining consistent researcher contact and offering regular check-ins, should be explored, but these alone may not be sufficient without addressing the broader ethical and structural barriers.

Despite these challenges, the study demonstrated strong staff engagement with the CTI model and high levels of consent to data linkage, suggesting that routine data could be a viable supplement for future evaluations. However, routine data is limited to 'hard outcomes' (eg healthcare use and criminal justice involvement) and may not capture the full impact of interventions like CTI on well-being or quality of life.

The findings highlight the need for systemic changes to support research in prison settings. These include revising NRC policies on incentives, streamlining approvals and investing in dedicated research teams familiar with the prison context. Future research should consider alternative trial designs that address ethical concerns around equipoise, perhaps by focusing on expected positive outcomes and personal autonomy rather than uncertainty of benefit. Additionally, the current pressures on HMPPS, including staffing shortages and policy churn, must be addressed to create a more stable environment for research and service delivery.

In conclusion, while the CTI model remains promising, this pilot trial underscores the practical and ethical challenges of conducting RCTs in this context. Future studies should prioritise flexible, context-sensitive designs, ethical engagement strategies and structural reforms to enable meaningful evaluation of interventions for prison-leavers experiencing homelessness.

**Author affiliations**
[1]Cardiff University, Cardiff, UK
[2]Centre for Trials Research, Cardiff University, Cardiff, UK
[3]Health Economics and Health Technology Assessment, University of Glasgow, Glasgow, UK
[4]University of Glasgow, Glasgow, UK
[5]Wrexham University, Wrexham, UK
[6]Heriot-Watt University, Edinburgh, UK
[7]MRC/CSO Social and Public Health Sciences Unit, University of Glasgow, Glasgow, UK
[8]School of Geography and Planning, Cardiff University, Cardiff, UK

**Acknowledgements** We would like to thank all CTI caseworkers and the intervention delivery organisation for working closely with us on this project along with all of our collaborators, as well as the Centre for Homelessness Impact who assisted with the development of participant-facing materials and appropriate language use. This study made use of anonymised data held in the Secure Anonymised Information Linkage (SAIL) Databank. We would like to acknowledge all the data providers who made anonymised data available for research via the SAIL Databank. Responsibility for the interpretation of the information supplied by SAIL is the authors' alone.

**Contributors** JL was the chief investigator and acted as guarantor. JL, MD, JW, RC-J, YM, PM, IM-J, SVK, SF and IT designed the study. ADNW created the first draft of this manuscript. All authors made substantive contributions to the development of the manuscript, critically reviewed and gave final approval to the manuscript. NJ

and ADNW conducted the process evaluation. ADNW and YM were responsible for the management of the trial. DG conducted the statistical analysis. MD and SOA developed and conducted the economic evaluation. Grammarly was used to assist with grammar and sentence structure.

**Funding** This research was supported by the National Institute for Health and Care Research (NIHR) Public Health Research Programme, grant number NIHR134281.

**Disclaimer** The views expressed are those of the authors and not necessarily those of HMPPS, the CTI delivery organisation, the NIHR or the Department of Health.

**Competing interests** None declared.

**Patient and public involvement** Patients and/or the public were involved in the design, or conduct, or reporting, or dissemination plans of this research. Refer to the Methods section for further details.

**Patient consent for publication** Not applicable.

**Ethics approval** This study involves human participants and was approved by Wales REC 3 (22/WA/0347). Participants gave informed consent to participate in the study before taking part.

**Provenance and peer review** Not commissioned; externally peer reviewed.

**Data availability statement** Data are available upon reasonable request. Following completion of the study, data will be made available upon request from the corresponding author. Individual level data from the test data linkage are not publicly available due to data sharing agreements limiting access to the research team only. Researchers wishing to use the SAIL Databank should direct their queries to the contact section of the SAIL webpage at https://saildatabank.com/contact/

**ORCID iDs**
Adam Dale Newman Williams https://orcid.org/0000-0002-4825-8997
Nina Jacob https://orcid.org/0000-0002-3240-4179
Detelina Grozeva https://orcid.org/0000-0003-3239-8415
Yvonne Moriarty https://orcid.org/0000-0002-7608-4699
Manuela Deidda https://orcid.org/0000-0002-0921-6970
Rebecca Cannings-John https://orcid.org/0000-0001-5235-6517
Srinivasa Vittal Katikireddi https://orcid.org/0000-0001-6593-9092
Jim Lewsey https://orcid.org/0000-0002-3811-8165

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
