## [Reviewer comments · BMJ Open]

ARTICLE DETAILS

Title (Provisional)

A pilot feasibility randomised controlled trial of a Critical Time Intervention for people leaving prison at risk of homelessness in England and Wales [PHaCT trial]

Authors

Williams, Adam Dale Newman; Jacob, Nina; Grozeva, Detelina; Lloyd, Barry; Moriarty, Yvonne; Deidda, Manuela; Achiaw, Samuel Owusu; Thomas, Ian; Lewis, Kelly; Cannings-John, Rebecca; Madoc-jones, Iolo; Fitzpatrick, Suzanne; Katikireddi, Srinivasa Vittal; Mackie, Peter; White, James; Lewsey, Jim

VERSION 1 - REVIEW

Reviewer	1
Name	Andersen, Lars
Affiliation	Rockwool Foundation Research Unit
Date	12-Feb-2025
COI	None

In this study, the authors describe a pilot trial of a critical time intervention for people at risk of experiencing homelessness upon release from prison. The aim of the study is to evaluate whether the intervention should be implemented at large scale (conclusion is that it should not). My comments are as follows:

(1) I applaud the authors for trying to handle such a multi-faceted and ambitious trial. This cannot be an easy task. The intervention seems well-intended and includes several (if not most) of important margins of the prison/homelessness challenge. This is no easy group to work with because their lives are often characterized by multiple and intertwined problems. The findings from the trial only show that we should add administrative/organizational challenges to the mix, which is valuable information.

(2) It is, however, unclear to me what we learn from this study that generalizes to the subject area or to practitioners working in the area. This is because so many margins of the trial went wrong that it is unclear whether the program was imperfect or whether challenges simply came from the range of implementation problems which the trial encountered. Is this about the intervention or about the structures surrounding the intervention? And if

problems arise because of the structures surrounding the intervention, would it not be better to try and change those structures to allow for large-scale implementation rather than just dropping the intervention?

(3) If one or a few barriers to implementation stood out, it would make sense to describe these in detail and explain to the research community what could be done to avoid such barriers moving forward. But there are so many barriers of such varied nature that seeing this as a study about which implementation barriers to be attentive to does not really make sense.

(4) Retention was super high, to the degree that the measurement of outcomes was pointless. Randomization was countermeasured by service providers. Treatment migration was a real threat (control group was offered other programs). Control groups could not be retained due to a lack of incentives. It was hard to track people because of administrative traits of the prisons and prison service. The probation service did not deliver as promised. There were so many problems in the lives of the target group, e.g., drug use exceeded 90 percent. The housing market on which the program relies was constrained. Either of these obstacles would challenge a trial such as this one – this trial encountered all of them. To me, this study seems more like a description of a failed intervention than a test of an interventions scalability.

Reviewer	2
Name	Lennox, Charlotte
Affiliation	University of Manchester, Manchester Academic Health Science Centre
Date	02-Sep-2025
COI	None

I am familiar with the authors work as I was also asked to review the process evaluation paper in relation to this study. The process evaluation required some major revision and the trial paper now needs to be reviewed in light of the changes to the process evaluation paper. As such, there is currently too much duplication between the papers and more clarity is needed to set out the unique aims and objectives of this paper.

Here I set out suggestions for each section.

Abstract

RCT in full in abstract

Useful in the abstract to include number of participants included and randomised and then the numbers to go with the % e.g. what is the n for 92%.

In the findings it says that recruitment progression was met but then in the limitations that the study it is highlighted that the study faced significant recruitment issues? This is confusing. Which is correct? I have highlighted this in the main paper too and as the authors to revise this.

Background

I am not sure the background adequately captures the UK landscape e.g. Shaw et al, did undertake a multisite trial, it did show impact of CTI on engagement with services and did provide an economic evaluation (albeit with some limitations). It was also possible to randomise people, and the trial design was acceptable – these are perhaps useful elements to add to enable contrasting with this study in the discussion section.

Research questions could be more clearly set out in addition to the main one about trial progression.

Methods

Line 123 – 129 – opportunities for concise writing e.g. 3 sentences in this paragraph all start with this pilot study etc.

Please add a justification for sample size target of 80, based on feasibility/pilot trials.

It might be useful to say that use of paper randomisation is limited to this study – most other prison trials do use an electronic system that is accessible in the prisons or randomisation is done off site. It is perhaps not a blanket issue, rather one that needed a work around.

Randomisation completed after baseline mentioned twice – could remove to be more concise. Useful to also have the exact date of 1st and last randomisation.

I wonder if the sections on data collection could split for baseline and then all follow-ups to provide clarity of the different processes for each.

Sentence not clear - Baseline data collection took place before randomisation within prison settings with follow -ups intending to be conducted at 150 probation offices, at the intervention delivery team's offices and over the telephone.

Why did you intend to use CRNs? And what were the logistical challenges?

Probation services were intended to help track people leaving 156 prison and follow -up data collection.? Who was doing the follow-ups? Was it still the CRNs in Wales and research team for England?

Did the follow-ups have windows? E.g. did 3 months have to happen at 3 months or between 2-4 months for example?

Line 178 – PhaCT not spelt out in full.

While it is useful to use the TIDieR to describe the intervention, it makes the content for this section very long and the use of the sub-headings does not really add anything (other than

extra words) suggest this section is condensed, but still able to capture the TIDieR information.

There is no mention of interviews in the data collection or who was interviewed?

Results

It would perhaps make more sense to move the first paragraph and incorporate into the method. I wonder if a table of some description is needed for the 4 sites which sets out who collected data, how CTI was delivered, how many interviews and observations and participants from each, what data was collected at each site. Within the narrative it is very hard to follow what data was collected when and who was involved and how the sites differed.

In the progression criteria, I understand that this is what may have been selected as recruitment criteria, but this is not actually the recruitment rate but the rate at which eligible participants consent to take part. This does not account for the very high rates of people not being eligible nor the lower-than-expected referral rates. Therefore, having this criteria as green but then going on to say that recruitment was an issue (which it was) is confusing. Suggest that the authors revise this to avoid the confusion.

Overall the results section contains far too much discussion and interpretation, which is perhaps best placed in the discussion section. Suggest the results section is revised to remove this and so just the results are presented. There is also significant overlap with the process evaluation paper. Suggest that this main trial paper now needs a significant rewrite to account for the changes to the PE paper. Suggest that the two papers need to be very distinct.

Retention – why could the eight participants returned to prison not be followed up? Why did 8 not provide contact details? Did they consent to being contacted, why could contact details not be obtained? For example, contact details from prison records.

Why was QoL decided as the indicative primary outcome? Please could a rationale be provided.

Indicative outcomes might be better presented in a table and by group (even with the limited number of participants), with average scores on the outcomes at baseline and follow-up.

Discussion

While QED are a valid option, they are not as robust as an RCT and would not show if an intervention works. RCTs are not unachievable within prisons, they are resource intensive and require a research team fully embedded in the prison, but they are, from experience, doable.

It is not mentioned until the recommendation section that the NRC did not allow incentives. I do not necessarily see that incentives is an issue just for the control group, there are other

way to support engagement with the control group, such as mainlining the same researcher, having text catch ups in between follow-ups etc. The NRC not allowing payment to compensate people for taking part in research, in the same way that the general population are able to do, is ethically questionable. I feel that this is the issue that needs to be discussed.

I look forward to reading the revised manuscript.

VERSION 1 - AUTHOR RESPONSE

Reviewer: 1

Dr. Lars Andersen, Rockwool Foundation Research Unit

Clearly articulate what the study contributes to the field despite the implementation challenges. Emphasise whether the findings reflect flaws in the intervention itself or in the surrounding systems and structures.	Discussion section fully revised to better address this comment.
Discuss how (or whether) the findings can inform future interventions or practice, given the extent of implementation issues. Consider whether structural changes could enable the intervention to succeed at scale.	Suggestion incorporated into revised discussion.
If possible, highlight one or two standout barriers to implementation. Provide detailed descriptions and practical recommendations for how these could be addressed in future studies or service deliver.	Suggestion incorporated into revised discussion.
Consider positioning the study as a case study of implementation failure or system-level barriers, rather than a test of intervention scalability.	Suggestion incorporated into revised discussion.
Avoid a binary conclusion (e.g. "should not be scaled"). Instead, reflect on what conditions would be necessary for the intervention to be viable.	Suggestion incorporated into revisions.

Reviewer: 2

Dr. Charlotte Lennox, University of Manchester

Abstract 1. Replace the abbreviation with “randomised controlled trial” on first mention. 2. Suggest the authors revise both the abstract and main paper for consistency.	1. RCT written fully and then abbreviated. 2. Abstract and main text have been revised to address all comments from reviewers.
Background 4. Expand the background to better reflect the UK landscape:  • Mention Shaw et al.’s multisite trial. • Highlight its findings on CTI’s impact on service engagement and economic evaluation. • Note that randomisation was feasible and acceptable in that study. 	4. Revised background section from comments provided.
5. Clearly state all research questions, not just the one about trial progression.	5. Revised to include clearer breakdown of the trial progression and outcomes.
Methods 6. Revise Lines 123–129: rephrase sentences to avoid repetition and improve flow. 7. Remove repeated mention of “randomisation completed after baseline” to streamline the text. 8. Clarify the unclear sentence about baseline and follow-up data collection to clearly distinguish settings and methods. 9. Explain why 80 participants were targeted. 10. Note paper-based randomisation was specific to this study due to contextual constraints, and contrast with typical electronic/off-site methods used in other prison trials. 11. Include the dates of the first and last randomisation to improve transparency. 12. Under data collection, separate descriptions for baseline and follow-up data collection to improve clarity.	6-8. Revised section to clarify based on comments 6-8. 9. Justification added to progression criteria section. 10. Revised section. 11. Dates added. 12. Section revised to separate baseline and follow-up for clarity.

13. Describe the rationale for involving Clinical Research Network (CRN) staff. 14. Specify who conducted follow-ups in Wales and England, CRNs or research team members. 15. State whether follow-ups had flexible time windows 16. Condense TIDieR Section 17. Include details about interviews in the data collection section.	13. Rationale for using CRNs added 14. Expanded on who conducted recruitment and follow-up. 15. Time windows for follow-up added. 16. Intervention description reduced. 17. Process evaluation description added to data collection section.
RESULTS 18. Relocate the opening paragraph of the Results section to the Methods section. 19. Create a table summarising 20. Revise the description of progression criteria to distinguish between recruitment rate and consent rate and resolve contradiction between “green” criteria and recruitment challenges 21. Reduce Discussion in Results and revise to minimise overlap with the process evaluation paper 22. Clarify Retention issues 23. Provide a rationale for selecting Quality of Life (QoL) as the indicative primary outcome. 24. Present Indicative Outcomes in a Table	Results section revised. 18. Moved section to the end of methods section. 19. Summarising table added, Table 2. 20. Added a sentence to address technically achieving the criteria versus not achieving the desired sample size. The criteria is named recruitment within the protocol so cannot be edited. 21. Discussion elements have been removed, and major elements of the evaluation have been removed. 22. Retention issues have been clarified. 23. Added to trial outcome measures. 24. Provided in supplemental material.
Discussion 25. Acknowledge that while quasi-experimental designs (QEDs) are valid, they are less robust than randomised controlled trials (RCTs) for determining intervention effectiveness. Emphasise that RCTs are feasible in prison settings,	Discussion was fully revised to match changes from results and address all comments from reviewers.

though resource-intensive, and require embedded research teams. 26. Move or expand the discussion on RCT feasibility to earlier in the section to strengthen the rationale for future trial design. 27. Introduce the National Research Committee (NRC) restriction on incentives earlier in the discussion, not just in the recommendations.  • Clarify that the lack of incentives affects all participants, not just the control group, and explore alternative engagement strategies (e.g. consistent researcher contact, regular check-ins). • Explicitly discuss the ethical concerns around the NRC's prohibition on compensating participants, especially given that such payments are standard in general population research. • Consider referencing equity and fairness in research participation. 	

VERSION 2 - REVIEW

Reviewer **2**

Name **Lennox, Charlotte**

Affiliation **University of Manchester, Manchester Academic Health
Science Centre**

Date **27-Oct-2025**

COI

Many thanks to the authors for dealing with the previous round of comments. I believe these have all been addressed and I have no further comments to make.